# Protein Language Model–Aligned Spectra Embeddings for De Novo Peptide Sequencing

## Abstract

We consider the problem of *de novo* peptide sequencing in tandem mass spectrometry, where the goal is to predict the underlying peptide sequence given a spectrum's fragment peaks and precursor information. We present PLMNovo, a constrained learning framework that leverages pre-trained protein language models (PLMs) to guide the training process. In particular, we cast peptide-spectrum matching as a constrained optimization problem that enforces alignment between spectrum and peptide embeddings produced by a spectrum encoder and a PLM, respectively. We use a Lagrangian primal-dual algorithm to train the spectrum encoder and the peptide decoder by solving the proposed constrained learning problem, while optionally fine-tuning the pre-trained PLM. Through numerical experiments on established benchmarks, we demonstrate that PLMNovo outperforms several state-of-the-art deep learning-based *de novo* sequencing algorithms.

## 1 Introduction

Tandem mass spectrometry (MS/MS) is central to bottom-up proteomics, wherein proteins are digested into peptides, fragmented, and recorded as spectra that capture sequence information (Neagu et al., 2022). The central computational problem is to translate each spectrum into its underlying peptide sequence (and often its modifications), enabling downstream protein inference, quantification, and biological interpretation. Accurate and scalable peptide identification underpins biomedical applications ranging from pathway mapping to biomarker and therapeutic development (Schirle et al., 2012; Liu et al., 2013; Pejchinovski et al., 2024). Yet, it remains challenging due to, among other reasons, noisy and incomplete fragmentation, instrument variability, and the sheer volume of data generated in modern experiments (McDonnell et al., 2022; Mao et al., 2023; Du et al., 2025).

Most production pipelines rely on *database search* methods: candidate peptides are enumerated from an *in silico*–digested protein database (with specified enzyme rules and optional variable modifications), theoretical fragment spectra are generated, and candidate matches are scored with procedures that enable false discovery rate (FDR) control (Kapp & Schütz, 2007). This strategy is robust when the database is appropriate and the search space is limited. However, it inherits several structural limitations. The primary challenge with database search methods is that they cannot identify peptides absent from the database, such as single-amino-acid variants or sequences from unmodeled organisms (van Puyenbroeck et al., 2025). Moreover, the computational burden grows combinatorially, e.g., with variable post-translational modifications (PTMs). As the candidate space expands, runtimes increase, scores become less discriminative, and FDR control becomes extremely challenging (Neuhauser, 2013).

*De novo* peptide sequencing removes the reliance on a fixed database by inferring sequences directly from spectra, casting the task as a structured supervised prediction problem under strict mass constraints. Recent deep learning approaches, ranging from transformer-based encoder–decoder models (Yilmaz et al., 2022; 2024; Eloff et al., 2025) to graph-based pipelines (Mao et al., 2023), have learned to incorporate precursor mass and charge to map spectra to amino acid sequences. Trained on large-scale spectral libraries and adaptable across instruments and fragmentation modes, these models can recover peptides not found in any database and reveal biology otherwise missed by search-based pipelines.

Most state-of-the-art deep learning-based *de novo* peptide sequencing pipelines create intermediate representations, or embeddings, of the fragment peaks contained in the spectrum, which are then used by a peptide decoder to reconstruct the underlying amino acid sequence. In essence, during training, this is a multi-modal learning problem, where we have access to two different modalities of the same data (i.e., spectrum peaks and peptide sequence). While prior efforts have been made to regularize this embedding space using both modalities (Jin et al., 2024), they have relied on peptide encoders that are trained from scratch and lack prior biological knowledge of the corresponding amino acid sequences, as compared to large-scale pre-trained biological foundation models.

In this work, we leverage *protein language models (PLMs)* (Bepler & Berger, 2021; Rives et al., 2021; Elnaggar et al., 2021; Lin et al., 2023; Hayes et al., 2025) to enhance the training procedure of deep learning *de novo* peptide sequencing pipelines. We propose a *constrained learning* formulation of *de novo* peptide sequencing, referred to as PLMNovo, where the spectra embeddings generated by the spectrum decoder are forced to be aligned with their corresponding peptide embeddings generated by a pre-trained PLM. We use a primal-dual training algorithm that identifies the right balance between minimizing the primary amino acid classification objective, while respecting the above *alignment constraints* between the spectra and peptide embeddings. By training our model over a massive dataset of 2 million peptide-spectrum matches (PSMs), we demonstrate the superiority of PLMNovo over state-of-the-art baselines. Moreover, we provide additional results that highlight the interplay between our proposed alignment constraints and the characteristics of peptides and spectra.

Our contributions are as follows:

- We propose, for the first time, a constrained learning pipeline that integrates pre-trained protein language models (PLMs) into the *de novo* peptide sequencing procedure.

- We provide a Lagrangian duality-based method for solving the sequencing problem under peptide-spectrum embedding alignment constraints.

- We numerically demonstrate that our proposed method outperforms state-of-the-art deep learning *de novo* sequencing pipelines across standard benchmarks.

## 2 RELATED WORK

### 2.1 *De Novo* PEPTIDE SEQUENCING

Building on the limitations of database search, *de novo* sequencing has become essential in settings where relevant peptide sequences are missing or incomplete, such as metaproteomics, immunopeptidomics, antibody sequencing, and paleoproteomics. Recent progress has been primarily driven by deep learning since the introduction of DeepNovo (Tran et al., 2017). Contemporary models span convolutional, transformer, and graph-based architectures. Representative examples include PointNovo (Qiao et al., 2021), Casanovo and its follow-up versions (Yilmaz et al., 2022; 2024; Melendez et al., 2024), PepNet (Liu et al., 2023), GraphNovo (Mao et al., 2023), InstaNovo (Eloff et al., 2025), and MassNet (Jun et al., 2025). Beyond accuracy, interpretability has begun to receive attention, with $\pi$-xNovo (Wang et al., 2024b) utilizing multi-head attention to link predicted residues to specific spectral peaks, thereby offering post-hoc explanations of model decisions.

Alongside methodological advances, the field has grappled with evaluation and reliability. Key open issues include principled false discovery rate (FDR) control that jointly considers database search and *de novo* results, understanding the tradeoff between database size and detection power, and estimating the fraction of "foreign" spectra in a dataset to contextualize performance gains. Benchmarking also requires care: retraining and hyperparameter tuning are often necessary for fair comparisons, as default settings can yield suboptimal results and distort conclusions (Zhou et al., 2024). Despite these challenges, rapid progress, the expansion of public datasets, and improvements in instrumentation suggest that *de novo* deep learning pipelines will continue to mature and broaden their impact in proteomics.

## 2.2 Protein Language Models

Large-scale language models for text have demonstrated that attention-based transformers (Vaswani et al., 2017) are powerful general-purpose sequence learners. The same recipe has been adopted for proteins: the availability of hundreds of millions of natural sequences (for example, from UniRef (Suzek et al., 2007; 2015) and BFD (Jumper et al., 2021)) enables pretraining at internet scale, turning protein sequences into the "language" on which to learn syntax (motifs), semantics (function), and long-range dependencies (contacts). This line of work has produced protein language models (PLMs) that learn rich contextual embeddings directly from amino-acid sequences and transfer surprisingly well to downstream biological tasks.

PLMs differ in architecture and objective but share the core idea of self-supervision. Masked language models, such as the ESM family (Rives et al., 2021; Lin et al., 2023; ESM Team, 2024; Hayes et al., 2025) and ProtT5 Elnaggar et al. (2021), learn to recover hidden residues from context and often scale to billions of parameters. Autoregressive generators, such as ProtGPT2 (Ferruz et al., 2022) and ProGen (Nijkamp et al., 2023), model next-token distributions and are used for controllable sequence generation. Moreover, context-based PLMs exploit multiple sequence alignment, or MSAs, that encode evolutionary variation explicitly (Rao et al., 2021; Truong Jr & Bepler, 2025; Akiyama et al., 2025). Recent efforts have extended the context length and improved efficiency (Chen et al., 2025a) and adopted diffusion-style generative processes for discrete sequences (Wang et al., 2024a).

The resulting representations from pre-trained PLMs have driven state-of-the-art or competitive performance across diverse applications, such as zero-shot and few-shot prediction of mutational effects (Meier et al., 2021; Brandes et al., 2023), functional annotation (Martínez-Redondo et al., 2025), and controllable protein and peptide design (Lee et al., 2024; Chen et al., 2025b). In proteomics specifically, PLM embeddings and priors have been utilized to enhance peptide property predictors, such as retention time, detectability, and MS/MS fragmentation pattern (Nakai-Kasai et al., 2025).

In this work, we present a novel application of PLMs in proteomics by integrating them into the *de novo* peptide sequencing pipeline, as described next.

## 3 Method

*De novo* peptide sequencing can be formulated as a supervised classification problem. Assume we have access to a set of $N$ annotated training samples $\{(\mathbf{p}_i, \mathbf{s}_i, \mathsf{prec}_i)\}_{i=1}^N$. For every $i \in \{1, \ldots, N\}$, $\mathbf{p}_i \in \mathcal{Y}^{L_i}$ represents the peptide sequence of length $L_i$, with $\mathcal{Y}$ denoting the amino acid alphabet, and $\mathbf{s}_i \in (\mathbb{R}_+ \times \mathbb{R}_+)^{K_i}$ denotes the observed spectrum composed of $K_i$ peaks, with each peak represented as a pair of $m/z$ (mass to charge ratio) and intensity values. Moreover, $\mathsf{prec}_i \in \mathbb{R}_+^3$ denotes the precursor information of the $i^{\text{th}}$ training sample, consisting of the precursor mass, precursor charge, and retention time. The goal of *de novo* peptide sequencing is to reconstruct the ground-truth peptide sequence given the observed spectrum and the precursor information. More precisely, we are interested in a parameterized function $f_\theta$ that solves the following empirical risk minimization problem:

$$\min_{\theta \in \Theta} \frac{1}{N} \sum_{i=1}^N \ell_{\mathsf{CE}}\Big(f_\theta(\mathbf{s}_i, \mathsf{prec}_i), \mathbf{p}_i\Big), \tag{1}$$

where $\ell_{\mathsf{CE}}(\cdot, \cdot)$ denotes the cross-entropy reconstruction loss function between the predicted and ground-truth peptide sequences, and $\Theta$ denotes the set of all possible model parameters. The supervised learning formulation in (1) has been used in the majority of the recent work on deep learning-based *de novo* peptide sequencing (Yilmaz et al., 2022; 2024; Eloff et al., 2025). Most of these studies break down the end-to-end function $f_\theta$ into an encoder $g_{\theta_{\mathsf{enc}}}$ and a decoder $h_{\theta_{\mathsf{dec}}}$, where the encoder maps the observed spectrum into intermediate peak-level representations, which are then used by the decoder, alongside the precursor information, to reconstruct the peptide sequence at the output, i.e.,

$$f_\theta(\mathbf{s}_i, \mathsf{prec}_i) = h_{\theta_{\mathsf{dec}}}\Big(g_{\theta_{\mathsf{enc}}}(\mathbf{s}_i), \mathsf{prec}_i\Big), \tag{2}$$

with the encoder and decoder parameterized using transformer architectures (Vaswani et al., 2017).

In this paper, we take a different approach from the supervised learning formulation in (1). Given the success of pre-trained protein language models (PLMs) in deriving informative representations from amino acid sequences, we hypothesized that including a PLM in the training process could benefit the generalization power of the *de novo* sequencing pipeline. More specifically, for a given peptide-spectrum match (PSM) $(\mathbf{p}, \mathbf{s})$, we propose *aligning* the peptide embedding generated by a PLM with the spectrum embedding $g_{\theta_{\text{enc}}}(\mathbf{s})$ created by the spectrum encoder.

For a peptide $p$ of length $L$, let $m_{\theta_{\text{PLM}}}(\mathbf{p}) \in \mathbb{R}^{L \times d'}$ denote the residue-level embeddings generated by a PLM $m_{\theta_{\text{PLM}}}$. Moreover, assume the peak-level embeddings generated by the spectrum encoder $g_{\theta_{\text{enc}}}$ lie in $R^d$, i.e., $g_{\theta_{\text{enc}}}(\mathbf{s}) \in \mathbb{R}^{K \times d}$ for a spectrum $\mathbf{s}$ with $K$ peaks. We then aggregate these embeddings using two pooling modules. To derive a unified spectrum-peptide co-embedding space, we further use a projection module to map the peptide embedding to $\mathbb{R}^d$. With a slight abuse of notation, we let $e_{\theta_{\text{s}}}(\cdot) \in \mathbb{R}^d$ denote the pooling function for the spectrum embeddings and $e_{\theta_{\text{p}}}(\cdot) \in \mathbb{R}^d$ represent the combined pooling and projection function for the peptide embeddings.

Our proposed method, PLMNovo, enforces the peptide-spectrum embedding alignment via a *constrained* learning approach (Chamon et al., 2022). Formally, we solve the following constrained optimization problem:

$$\min_{\theta_{\text{enc}}, \theta_{\text{dec}}, \theta_{\text{s}}, \theta_{\text{p}}, \theta_{\text{PLM}}} \frac{1}{N} \sum_{i=1}^{N} \ell_{\text{CE}} \left[ h_{\theta_{\text{dec}}} \left( g_{\theta_{\text{enc}}}(\mathbf{s}_i), \text{prec}_i \right), \mathbf{p}_i \right], \tag{3a}$$

$$\text{s.t.} \quad \left\| e_{\theta_{\text{s}}} \left( g_{\theta_{\text{enc}}}(\mathbf{s}_i) \right) - e_{\theta_{\text{p}}} \left( m_{\theta_{\text{PLM}}}(\mathbf{p}_i) \right) \right\|_2^2 \leq \epsilon, \quad \forall i \in \{1, \ldots, N\}, \tag{3b}$$

where $\|\mathbf{x} - \mathbf{y}\|_2$ denotes the Euclidean distance between two vectors $\mathbf{x}, \mathbf{y} \in \mathbb{R}^d$. The learning problem in (3) attempts to find the model parameters that not only minimize the primary *de novo* sequencing objective in (3a), but also ensure that the spectra and peptide embeddings are closely aligned. The alignment is introduced as a *per-PSM constraint* in (3b), where the squared Euclidean distance between the aggregated spectrum and peptide embeddings, both in $\mathbb{R}^d$, is at most $\epsilon$. The upper bound, $\epsilon$, is treated as a hyperparameter: extremely low values of $\epsilon$ could make the problem infeasible or lead to degenerate solutions (where all embeddings collapse to a small subset of the embedding space), while $\epsilon \to \infty$ reverts the problem to the original unconstrained problem in (1). Therefore, its choice is critical in generating informative spectrum (and peptide) embeddings. Figure 1 illustrates an overview of PLMNovo.

Observe that the optimization variables in (3) include those of the spectrum encoder $\theta_{\text{enc}}$, the peptide decoder $\theta_{\text{dec}}$, the spectrum pooling $\theta_{\text{s}}$, the peptide pooling and projection $\theta_{\text{p}}$, and the PLM $\theta_{\text{PLM}}$. While all these parameters are involved during the training phase, only the spectrum encoder and peptide decoder parameters (i.e., $\theta_{\text{enc}}, \theta_{\text{dec}}$) are used during inference. Furthermore, the PLM parameters can be kept frozen (at a pre-trained checkpoint), or trained end-to-end (e.g., via fine-tuning (Hu et al., 2022; Schmirler et al., 2024; Sledzieski et al., 2024)). We note that such an MSE-based alignment approach resembles the BYOL self-supervised learning method (Grill et al., 2020), but instead of augmentations, here we consider embeddings from two different data modalities and corresponding feature extractors.

### 3.1 Primal-Dual Training

To solve the constrained learning problem in (3), we move to the dual domain (Boyd & Vandenberghe, 2004) and write the Lagrangian function as

$$\mathcal{L}(\theta_{\text{enc}}, \theta_{\text{dec}}, \theta_{\text{s}}, \theta_{\text{p}}, \theta_{\text{PLM}}, \boldsymbol{\lambda})$$

$$= \frac{1}{N} \sum_{i=1}^{N} \ell_{\text{CE}} \left[ h_{\theta_{\text{dec}}} \left( g_{\theta_{\text{enc}}}(\mathbf{s}_i), \text{prec}_i \right), \mathbf{p}_i \right] + \sum_{i=1}^{N} \lambda_i \left[ \left\| e_{\theta_{\text{s}}} \left( g_{\theta_{\text{enc}}}(\mathbf{s}_i) \right) - e_{\theta_{\text{p}}} \left( m_{\theta_{\text{PLM}}}(\mathbf{p}_i) \right) \right\|_2^2 - \epsilon \right], \tag{4}$$

where $\lambda_i \geq 0$ is the *Lagrangian dual multiplier* corresponding to the $i^{\text{th}}$ training sample, and $\boldsymbol{\lambda} \in \mathbb{R}_+^N$ denotes the vector of all dual multipliers. The dual of problem (3) is then given by the saddle-point problem

$$\max_{\boldsymbol{\lambda} \in \mathbb{R}_+^N} \min_{\theta_{\text{enc}}, \theta_{\text{dec}}, \theta_{\text{s}}, \theta_{\text{p}}, \theta_{\text{PLM}}} \mathcal{L}(\theta_{\text{enc}}, \theta_{\text{dec}}, \theta_{\text{s}}, \theta_{\text{p}}, \theta_{\text{PLM}}, \boldsymbol{\lambda}). \tag{5}$$

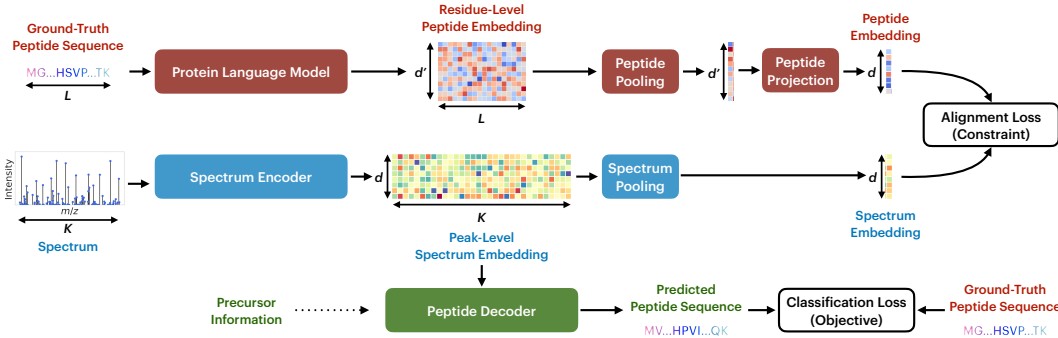

Figure 1: Our proposed architecture, PLMNovo, consists of an encoder-decoder pair, whose intermediate embeddings are constrained by a protein language model (PLM). In particular, the spectrum fragment peak information (comprising $K$ ($m/z$, intensity) pairs) is mapped to a peak-level $K \times d$ spectrum embedding using a spectrum encoder. These intermediate embeddings are then fed into a peptide decoder, alongside the precursor mass and charge, to predict the corresponding peptide sequence. The predicted peptide sequence is compared to the ground-truth peptide sequence using the cross-entropy loss, which constitutes the primary objective function. Simultaneously, the ground-truth peptide sequence (comprising $L$ amino acids) is mapped to a residue-level $L \times d'$ peptide embedding using a PLM. The spectrum and peptide embeddings are aggregated using two pooling modules, and the aggregated peptide embedding is projected to the same embedding space as the spectrum embedding (i.e., $\mathbb{R}^d$). The squared Euclidean distance between the resulting peptide and spectrum embeddings is then enforced to be bounded by a constant, which acts as a per peptide-spectrum match (PSM) constraint in the optimization problem. While the spectrum encoder-decoder pair, as well as the pooling and projection modules, are trained end-to-end, the PLM is fine-tuned from a pre-trained checkpoint via the alignment loss gradients.

To solve this problem, we can use a primal-dual approach (Boyd & Vandenberghe, 2004; Fioretto et al., 2021; Elenter et al., 2022), where we alternate between updating the model parameters and the dual multiplier. More specifically, in each primal iteration, the model parameters are updated using *gradient descent* on the Lagrangian, i.e.,

$$\boldsymbol{\theta} \leftarrow \boldsymbol{\theta} - \eta_{\boldsymbol{\theta}} \frac{\partial \mathcal{L}}{\partial \boldsymbol{\theta}}, \tag{6}$$

where $\eta_{\boldsymbol{\theta}}$ denotes the primal learning rate, and to ease the notation, we aggregate all the model parameters into $\boldsymbol{\theta} = [\theta_{\mathsf{enc}}, \theta_{\mathsf{dec}}, \theta_{\mathsf{s}}, \theta_{\mathsf{p}}, \theta_{\mathsf{PLM}}]$. Then, in each dual iteration, the dual multipliers are updated using *projected gradient ascent* on the Lagrangian, i.e.,

$$\boldsymbol{\lambda} \leftarrow \left[ \boldsymbol{\lambda} + \eta_{\boldsymbol{\lambda}} \frac{\partial \mathcal{L}}{\partial \boldsymbol{\lambda}} \right]_+, \tag{7}$$

where $\eta_{\boldsymbol{\lambda}}$ denotes the dual learning rate, and $[\cdot]_+ := \max(\cdot, 0)$ represents elementwise mapping onto the $\mathbb{R}^N_+$. Combining (4) and (7), we can rewrite the update of each Lagrangian multiplier in closed form as

$$\lambda_i \leftarrow \left[ \lambda_i + \eta_{\boldsymbol{\lambda}} \left( \left\| e_{\theta_{\mathsf{s}}}\left( g_{\theta_{\mathsf{enc}}}(\mathbf{s}_i) \right) - e_{\theta_{\mathsf{p}}}\left( m_{\theta_{\mathsf{PLM}}}(\mathbf{p}_i) \right) \right\|_2^2 - \epsilon \right) \right]_+. \tag{8}$$

The closed-form update in (8) implies that the dual multiplier corresponding to each PSM accumulates the alignment constraint violations over the course of the training process. In other words, if for a PSM, the spectra and peptide embeddings are perfectly aligned, its corresponding dual multiplier remains at zero, whereas in the case of a PSM for which there is a significant misalignment between the spectra and peptide embeddings, its dual multiplier keeps increasing. This shows how the proposed constrained formulation adapts the importance of different training sample alignments by dynamically adjusting the importance of each PSM in the Lagrangian in (4).

# 4 NUMERICAL RESULTS

## 4.1 EXPERIMENTAL SETTINGS

We base PLMNovo's implementation on Casanovo 4.2 (Melendez et al., 2024), a state-of-the-art autoregressive *de novo* sequencing pipeline trained on a dataset of 2 million PSMs, all of which are digested using trypsin, the standard enzyme used in tandem mass spectrometry (Melendez, 2024). We train all PLMNovo models on this dataset. This dataset originates from the MassIVE Knowledge Base (Wang et al., 2018) and is accompanied by a test split containing 200,000 tryptic samples. In what follows, we refer to this dataset as the MSKB dataset. Casanovo 4.2 is built on the previous versions of Casanovo (Yilmaz et al., 2022; 2024). We follow the exact hyperparameters used by Melendez et al. (2024), including using a 9-layer transformer-based encoder-decoder architecture with a $d = 512$-dimensional embedding space, and a beam search decoding mechanism with $k = 10$ beams. The rest of the hyperparameters and model details are mentioned in Appendix A. We retrain Casanovo v4.2 on the MSKB training set, and all results reported below corresponding to this architecture are based on our retrained version.

As for the PLMs, we use two PLM architectures from the ESM-2 family (Lin et al., 2023), namely the 8M (with $d' = 320$) and 650M (with $d' = 1280$) versions. We remove any post-translational modifications (PTMs) from the ground-truth peptide sequences before feeding them to the PLMs to respect their token vocabularies, which are based on the canonical amino acids. To manage computational complexity, we leverage low-rank adaptation, or LoRA (Hu et al., 2022), to fine-tune the PLM parameters, focusing on only the key and value parameter matrices in the self-attention layers, as recommended in prior work (Sledzieski et al., 2024). We utilize average pooling to aggregate both peptide and spectrum embeddings.

We perform grid search on two important hyperparameters: the alignment constraint bound ($\epsilon \in \{10^{-1}, 10^{-2}, 10^{-3}\}$) in (3b), and the LoRA fine-tuning rank ($r \in \{0, 2, 4\}$). Treating the MSKB test set as a validation split, for each PLM, we select the $(\epsilon, r)$ combination that leads to the lowest amino acid classification loss. Our implementation code can be found at `https://github.com/AnonMS2/PLMNovo`.

## 4.2 PERFORMANCE ON THE MSKB AND MULTI-ENZYME TEST SETS

The top portion of Table 1 compares the performance of PLMNovo and Casanovo v4.2 on the MSKB test set. Across all amino acid-level and peptide-level metrics, PLMNovo, especially with the 8M version of ESM-2, outperforms the base Casanovo v4.2 model, demonstrating the performance boost that the PLM-guided alignment constraints provide in PLMNovo.

While the MSKB test set provides an in-distribution evaluation setting, we also tested our models on a held-out non-tryptic multi-enzyme dataset (Melendez, 2024; Melendez et al., 2024). As the bottom portion of the Table 1 shows, PLMNovo also outperforms Casanovo v4.2 on this out-of-distribution dataset. Interestingly, ESM-2 650M significantly outperforms ESM-2 8M on this dataset, suggesting that the smaller-scale PLM may have led to slight overfitting of PLMNovo on tryptic data.

| Dataset | Method | Classification Loss (↓) | AA Precision (↑) | AA Recall (↑) | Peptide Precision (↑) |
|---|---|---|---|---|---|
| **MSKB (Tryptic)** | Casanovo v4.2 (Melendez et al., 2024) | 0.1983 | 0.8919 | 0.8884 | 0.7381 |
| | PLMNovo (ESM-2 8M) | **0.1919** | 0.8973 | **0.8921** | **0.7411** |
| | PLMNovo (ESM-2 650M) | 0.1941 | **0.8976** | 0.8915 | 0.7400 |
| **Multi-Enzyme (Non-Tryptic)** | Casanovo v4.2 (Melendez et al., 2024) | 0.9949 | 0.5200 | 0.5211 | 0.2467 |
| | PLMNovo (ESM-2 8M) | 0.9723 | 0.5249 | 0.5283 | 0.2410 |
| | PLMNovo (ESM-2 650M) | **0.9714** | **0.5356** | **0.5323** | **0.2483** |

Table 1: Amino acid-level and peptide-level performance comparison on the MSKB and multi-enzyme test sets (Melendez, 2024). Numbers in **bold** represent the best result under each (metric, dataset) combination.

### 4.3 PERFORMANCE ON THE NINE-SPECIES BENCHMARK

We next evaluated PLMNovo, pre-trained on the MSKB training set, on the nine-species benchmark, which is a standard dataset used by the majority of prior work on *de novo* peptide sequencing. As Table 2 shows, while PLMNovo provides competitive performance in terms of peptide recall, it outperforms all other baselines in terms of the amino acid precision.

| Metric | Method | Species | | | | | | | | | Average |
|---|---|---|---|---|---|---|---|---|---|---|---|
| | | *Bacillus* | *C. bacteria* | *Honeybee* | *Human* | *M. mazei* | *Mouse* | *Ricebean* | *Tomato* | *Yeast* | |
| AA Precision (↑) | DeepNovo (Tran et al., 2017) | 0.742 | 0.602 | 0.630 | 0.610 | 0.694 | 0.623 | 0.679 | 0.731 | 0.750 | 0.673 |
| | PointNovo (Qiao et al., 2021) | 0.768 | 0.589 | 0.644 | 0.606 | 0.712 | 0.626 | 0.730 | 0.733 | 0.779 | 0.687 |
| | Casanovo (Yilmaz et al., 2022) | 0.749 | 0.603 | 0.629 | 0.586 | 0.679 | 0.689 | 0.668 | 0.721 | 0.684 | 0.667 |
| | AdaNovo (Xia et al., 2024) | 0.739 | 0.642 | 0.650 | 0.618 | 0.728 | 0.646 | 0.719 | 0.740 | **0.793** | 0.697 |
| | Casanovo v2 (Yilmaz et al., 2024) | 0.790 | 0.681 | 0.706 | **0.676** | **0.755** | **0.760** | 0.748 | 0.785 | 0.752 | 0.739 |
| | Casanovo v4.2 (Melendez et al., 2024) | 0.793 | 0.678 | 0.705 | 0.668 | 0.687 | 0.756 | 0.753 | 0.791 | 0.768 | 0.733 |
| | PLMNovo (ESM-2 8M) | 0.794 | 0.684 | 0.707 | 0.670 | **0.755** | 0.759 | 0.764 | 0.790 | 0.755 | 0.742 |
| | PLMNovo (ESM-2 650M) | **0.797** | **0.688** | **0.709** | **0.676** | **0.755** | **0.760** | **0.766** | **0.795** | 0.771 | **0.746** |
| Peptide Recall (↑) | DeepNovo (Tran et al., 2017) | 0.449 | 0.253 | 0.330 | 0.293 | 0.422 | 0.286 | 0.436 | 0.454 | 0.462 | 0.376 |
| | PointNovo (Qiao et al., 2021) | 0.518 | 0.298 | 0.396 | 0.351 | 0.478 | 0.355 | 0.511 | 0.513 | 0.534 | 0.439 |
| | CasaNovo (Yilmaz et al., 2022) | 0.537 | 0.330 | 0.406 | 0.341 | 0.478 | 0.426 | 0.506 | 0.521 | 0.490 | 0.448 |
| | AdaNovo (Xia et al., 2024) | 0.528 | 0.372 | 0.431 | 0.373 | 0.496 | 0.467 | 0.546 | 0.530 | 0.593 | 0.481 |
| | Casanovo v2 (Yilmaz et al., 2024) | **0.622** | **0.446** | **0.493** | **0.446** | **0.557** | **0.483** | **0.589** | **0.618** | **0.599** | **0.539** |
| | Casanovo v4.2 (Melendez et al., 2024) | 0.603 | 0.421 | 0.478 | 0.437 | 0.498 | 0.468 | 0.558 | 0.608 | 0.584 | 0.517 |
| | PLMNovo (ESM-2 8M) | 0.602 | 0.423 | 0.484 | 0.434 | 0.544 | 0.470 | 0.565 | 0.606 | 0.571 | 0.522 |
| | PLMNovo (ESM-2 650M) | 0.601 | 0.425 | 0.483 | 0.432 | 0.544 | 0.464 | 0.576 | 0.606 | 0.579 | 0.523 |

Table 2: Performance comparison results in terms of amino acid precision and peptide recall on the nine-species dataset (Tran et al., 2017). The performance of the baseline methods, except for Casanovo v4.2, is reported from (Zhang et al., 2025b). **Bolded** and underlined results represent the first and second best performance per species (or averaged across species in the last column), respectively.

### 4.4 IMPACT OF THE CONSTRAINTS ON THE EMBEDDING SPACE

Figure 2-(a) provides a two-dimensional visualization of the embedding space occupied by the peptide and spectrum embeddings of the MSKB test set using t-SNE (Maaten & Hinton, 2008) under different alignment constraint bounds. As the figure shows, our proposed alignment constraints highly regularize the peptide-spectrum co-embedding space, where matching peptide and spectrum embeddings lie close to each other. This is in contrast to the unconstrained case ($\epsilon \to \infty$), where peptides and spectra embeddings are very far from each other.

Furthermore, as shown in Figure 2-(b), tighter constraint bounds generally lead to lower Euclidean distances for the PSMs in the embedding space, even though enforcement of such alignment gets more and more challenging as $\epsilon$ is reduced.

### 4.5 INTERPRETATION OF EMBEDDING ALIGNMENT

**Alignments for Different Peptide and Spectrum Scales.** Figures 3-(a) and 3-(b) show the squared Euclidean distance between peptide and spectrum embeddings for different peptide lengths and number of spectrum peaks. Interestingly, the PSM co-embedding distance increases for extremely short or long peptides, and decreases with an increasing number of peaks. The latter phenomenon is intuitive, since having more peaks can lead to a more informative representation of the spectrum, hence improving the peptide decoding process. The former phenomenon has also been previously reported in the literature (Zhou et al., 2024), where longer peptide sequences are generally more complicated to decode, especially when using autoregressive methods, and shorter peptide sequences suffer from a lack of sufficient training data. The inclusion of more sophisticated pooling methods for spectrum and peptide embeddings (Zhang et al., 2020; Stärk et al., 2021; NaderiAlizadeh & Singh, 2025; Amir & Dym, 2025; Tartici et al., 2025; NaderiAlizadeh et al., 2025)), as well as non-autoregressive decoding algorithms (Zhang et al., 2025a;b)), could enhance the performance of PLMNovo, especially for longer peptide sequences.

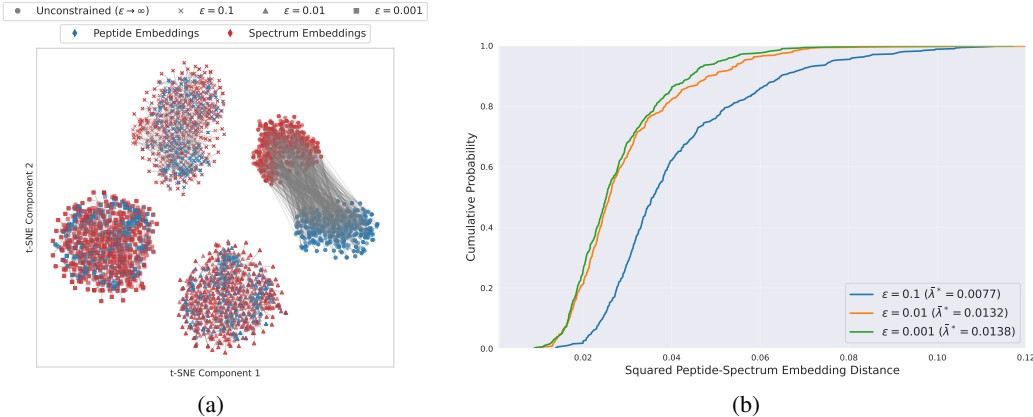

Figure 2: (a) t-SNE (Maaten & Hinton, 2008) visualization of the peptide-spectrum co-embedding space produced by PLMNovo at various alignment constraint bounds ($\epsilon$) levels on a subset of 1000 PSMs from the MSKB test set. (b) Empirical cumulative distributions of the peptide-spectrum embedding distances under different constraint levels, alongside the corresponding optimal Lagrangian dual multipliers, averaged across the training samples.

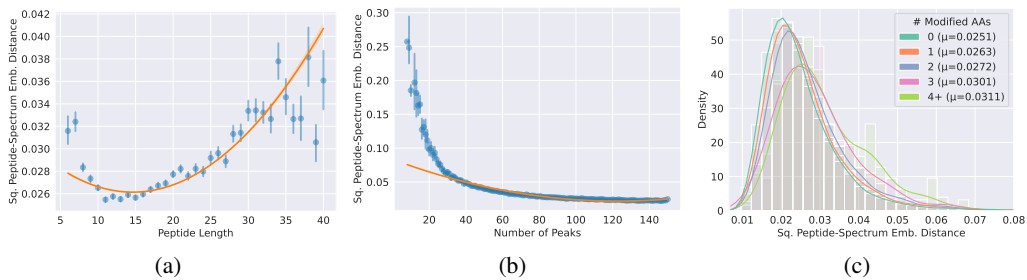

Figure 3: Squared Euclidean distance of PSM embeddings vs. (a) peptide length and (b) number of peaks in the spectrum. (c) Distribution of the squared Euclidean PSM embedding distance for different numbers of modified amino acids in the peptide sequences.

**Impact of Post-Translational Modifications.** Post-Translational Modifications, or PTMs, are biological processes where proteins are chemically modified after translation. While prevalent in proteomics, modified amino acids resulting from PTMs, such as oxidation of methionine and deamidation of asparagine or glutamine, constitute a small fraction of the amino acids in tandem mass spectrometry training datasets. This has led to previous de novo sequencing models struggling to decode PTMs, thereby motivating methods especially designed to handle modified residues, such as AdaNovo (Xia et al., 2024) and PrimeNovo (Zhang et al., 2025a). We sought to understand the impact of PTMs on PLMNovo's learned co-embedding space. Figure 3-(c) shows the histograms of squared peptide-spectrum embedding distances for peptide sequences with different numbers of modified amino acids on the MSKB test set. As the figure demonstrates, the embedding gap between a spectrum and its corresponding peptide increases with the number of PTMs in the peptide sequence. We hypothesize that this limitation stems from the inability of common PLMs, such as the ESM-2 family, to handle PTMs. PLM architectures specifically designed to handle PTM tokens, such as PTM-Mamba (Peng et al., 2025), can potentially bridge this gap and enhance PLMNovo's ability to identify modified amino acids during the decoding process.

## 4.6 ABLATION STUDY

Figure 4 presents an ablation study of PLMNovo in terms of different alignment constraint bounds and PLM fine-tuning ranks for 8M and 650M versions of ESM-2. As the figure shows, fine-tuning the PLM generally helps improve PLMNovo's performance compared to keeping the PLM frozen

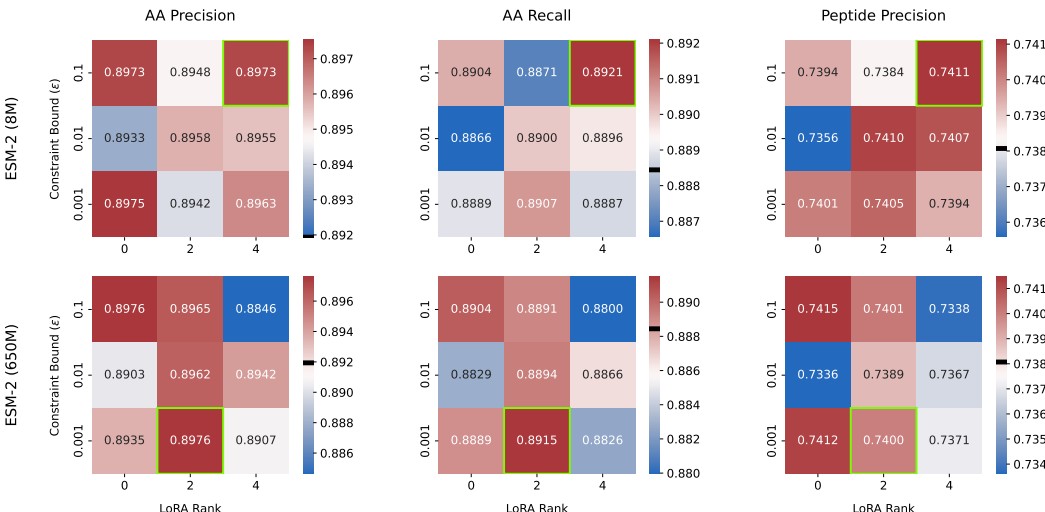

Figure 4: Ablation study of PLMNovo's performance on the MSKB test set in terms of the alignment constraint bound ($\epsilon$) in (3b), as well as the fine-tuning rank of the PLM, where a rank of zero implies that the PLM was frozen. Boxes highlighted in green correspond to the selected hyperparameter combination. Furthermore, the black lines in the colorbars represent Casanovo 4.2's performance, which is equivalent to PLMNovo's unconstrained performance (i.e., $\epsilon \to \infty$).

(LoRA rank = 0). While the majority of the configurations in our grid search exceed the performance of Casanovo v4.2, these results show that tuning these hyperparameters is critical to maximize PLM-Novo's predictive power.

## 5 DISCUSSION AND CONCLUDING REMARKS

We presented PLMNovo, a novel deep learning method for *de novo* peptide sequencing, which enforces the proximity of matching peptides and spectra in a co-embedding space. While the spectrum embeddings are generated using a spectrum encoder, we use pre-trained protein language models (PLMs) to create the peptide embeddings. We formulate the sequencing problem using a constrained learning framework and adopt a primal-dual algorithm to train the spectrum encoder and peptide decoder, while fine-tuning the PLM. Numerical experiments demonstrated that PLMNovo surpasses other baseline deep learning *de novo* peptide sequencing methods in various benchmarks.

In the future, we envision several immediate enhancements to PLMNovo, such as investigating other PLM architectures and families (e.g., (Elnaggar et al., 2021; Nijkamp et al., 2023; ESM Team, 2024; Truong Jr & Bepler, 2025; Peng et al., 2025)) and alternative embedding pooling strategies (e.g., (Tartici et al., 2025; NaderiAlizadeh & Singh, 2025)). While our pipeline is implemented based on Casanovo 4.2 (Melendez et al., 2024), we expect our gains to transfer to more recent *de novo* sequencing pipelines that employ techniques such as curriculum learning (Zhang et al., 2025b), sequence re-ranking (Qiu et al., 2025), missing fragmentation imputation (Du et al., 2025), and non-autoregressive decoding (Zhang et al., 2025a). Finally, extension of the proposed constrained learning approach to data-independent acquisition (DIA) (Sanders et al., 2025), a more challenging protocol than the data-dependent acquisition (DDA) setting studied in this work, is an essential avenue for further impact of PLMNovo in mass spectrometry research.

## 6 USE OF LARGE LANGUAGE MODELS

Large Language Models (LLMs) were utilized to aid in manuscript review and editorial refinement.

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

## A  ADDITIONAL PLMNOVO IMPLEMENTATION DETAILS

**Architecture.**  Similar to Casanovo 4.2 (Melendez et al., 2024), PLMNovo utilizes transformers (Vaswani et al., 2017) to map spectral information onto amino acid sequences via an encoder-decoder architecture. Peak characteristics ($m/z$ ratios and intensities) undergo distinct encoding procedures, with sinusoidal functions for mass values and learnable linear mappings for intensities, producing unified high-dimensional representations through summation. The encoder component processes these peak embeddings using multi-head attention to establish inter-peak relationships and contextual understanding across the entire spectrum. The peak embeddings then guide the decoder's sequential amino acid prediction task.

Sequence generation operates through step-wise autoregressive decoding initiated with precursor information. In particular, precursor mass and charge values undergo a sinusoidal transformation and a linear layer, respectively, before being integrated into unified embeddings. The decoder leverages both spectral context and precursor information to initiate the construction of amino acid sequences. At each decoding step, the decoder processes embeddings corresponding to the precursor characteristics and all previously predicted amino acids. Beam search maintains diversity by tracking the top $k$ scoring hypotheses throughout decoding, expanding sequences until natural termination or mass constraint violation occurs, with the output sequence being the maximum-scoring complete sequence. Quality control applies mass accuracy filters, penalizing predictions whose theoretical precursor masses exceed specified deviation limits (50 ppm threshold) from observed values.

**Hyperparameters.**  The length of predicted peptides is set to be between a minimum of 6 and a maximum of 100 amino acids. We select at most 150 peaks within each spectrum, with a minimum intensity of 0.01, and a $m/z$ ratio between 50 and 2500. We use a batch size of 32 during training and train the model for 7 epochs (Melendez et al., 2024). We use the Adam optimizer with a primal learning rate of $\eta_{\boldsymbol{\theta}} = 5 \times 10^{-4}$ and a weight decay of $10^{-5}$ and a dual learning rate of $\eta_{\boldsymbol{\lambda}} = 10^{-2}$. The primal learning rate is warmed up linearly in the first $10^5$ iterations, followed by a cosine-shaped decay with a half period of $6 \times 10^5$ iterations. A label smoothing factor of $10^{-2}$ is used when calculating the training classification loss. The encoder and decoder each have 9 self-attention layers, each containing 8 attention heads, with a latent representation dimension of 512 and a fully-connected layer dimension of 1024. For LoRA, we set $\alpha = 4 \times r$ for any selected rank $r$ (Sledzieski et al., 2024).