# OpenReview forum: "Protein Language Model–Aligned Spectra Embeddings for De Novo Peptide Sequencing"
_ICLR.cc/2026/Conference — ICLR 2026 Conference Withdrawn Submission_

### Official Review · Reviewer_Ft2g · 2025-10-21

**Soundness:** 1
**Presentation:** 3
**Contribution:** 1
**Rating:** 2
**Confidence:** 4

**Summary:**

This paper proposes a denovo peptide sequencing method using PLM (protein pretrain model) for feature alignment, allowing the peptide decoder of the model to acquire knowledge from PLM rather than undergoing training from scratch. The experiments in this paper are based on casanovo 4.2 + ESM 2;

**Strengths:**

1. Utilizing the prior knowledge of large-scale pre-trained models for transfer learning or few-shot learning sounds like a sensible approach; applying this approach to de novo should also be a novel idea;

**Weaknesses:**

1. The motivation of the article is not sound; it states: "While prior efforts have been made to regularize this embedding space using both modalities (Jin et al., 2024), they have relied on peptide encoders that are trained from scratch and lack prior biological knowledge of the corresponding amino acid sequences, as compared to large-scale pre-trained biological foundation models." However, there is no reasonable biological knowledge or experimental verification to indicate that de novo models indeed require the prior knowledge of PLMs. PLMs are generally pre-trained on protein sequences (primary structure), while the sequences output by de novo can be regarded as a part of proteins, which are peptides; meanwhile, there is no experiment or evidence to suggest that sequencing tasks and protein representation tasks are related. The premise for the knowledge learned from pre-training to be helpful for downstream tasks is that their task forms are related and the domains are similar, which is not reflected in this article;

2. The performance improvement is negligible; note that PLMNovo is trained based on casanovo 4.2+ESM2, but from Tables 1 and 2, the improvement is very minor compared with casanovo 4.2, and on the peptides in Table 2, it is even inferior to the earlier casanovo 2; if the performance improvement based on casanovo 4.2 is too small, it indicates that the method may just be a trick;

3. The ablation experiment presented in Figure 4 seems to demonstrate the above point. Under the same lora rank, the performance of the model is minimally affected by the Constraint Bound (by no more than 0.003), indicating that the Constraint Bound barely plays any role. This also implies that aligning spectra embedding and peptide embedding (the core method proposed in the article) has little effect.

4. The differences brought by using the 8M and 650M versions of ESM are minimal (Tables 1 and 2), indicating that the knowledge from ESM indeed has little effect on the denovo task;

5. Various experimental details are questionable:
    1. The article seems to use a new MSKB dataset, but NovoBench, which is commonly used in the AI community, is not involved;
    2. Beam search=10 was used, but it is not stated whether other baselines also incorporated beam search and underwent re-experimentation. The impact of varying beam search sizes can be significant (based on the experience of the NLP community);
    3. It is not clearly explained how the framework of the model differs between the train and infer stages. This is important in this article because the infer stage lacks peptide sequence (label) and ESM
    4. Line 326 mentions 'We next evaluated PLMNovo, pre-trained on the MSKB training set, on the nine-species benchmark.' It seems that other baselines have not undergone such pre-training (nor even mentioned the MSKB dataset in other paper), so this comparison is unfair."
    5. Section 4.5 of the article mentions that PTM can bring negative impacts. However, as far as I know, post-translational modifications of peptide sequences are quite common, indicating that this article may underestimate the negative impacts of PTM Novo in practical usage scenarios;

In summary, this paper presents a novel and intriguing idea. Unfortunately, there are significant issues with both the motivation and experimental verification, rendering the core innovation of this paper, "a constrained learning framework that leverages pre-trained protein language models (PLMs) to guide the training process," is not solid and reasonable. This seems more like an experimental attempt than a significant innovation.

**Questions:**

in “Weaknesses”

---

### Official Review · Reviewer_XYhM · 2025-10-28

**Soundness:** 2
**Presentation:** 2
**Contribution:** 1
**Rating:** 2
**Confidence:** 3

**Summary:**

This paper proposes PLMNovo, a framework for de novo peptide sequencing based on constrained learning that enforces alignment between spectrum embeddings and PLM-derived peptide embeddings via a per-PSM Lagrangian/primal-dual scheme. And reports modest improvements over baselines on several benchmarks.

**Strengths:**

- The core motivation of leveraging rich, pre-trained representations from Protein Language Models (PLMs) to guide the training of a spectrum encoder is reasonable.
- The paper proposes a constrained learning framework based on a Lagrangian primal-dual optimization algorithm to enforce this alignment between spectrum and peptide embeddings.

**Weaknesses:**

Major
- Limited Novelty: The contribution appears incremental. The core concept of aligning spectrum and peptide embeddings is highly similar to ContraNovo (Jin et al., 2024). The primary distinction, using a pre-trained PLM versus a from-scratch encoder, is an incremental modification, not a foundational leap。
- Missing comparisons to relevant methods (e.g., ContraNovo, pi-PrimeNovo, InstaNovo), which is essential given the claimed contribution.In particular, given that ContraNovo adopts a contrastive regularization between spectrum and peptide spaces, a direct head-to-head comparison is essential to support claims of improvement attributable to the use of a pretrained PLM.

**Questions:**

- Why was the MSKB test split used for hyperparameter selection rather than a held-out validation set? Please clarify whether it could introduce data leakage in the reported results.

---

### Official Review · Reviewer_tDiv · 2025-10-28

**Soundness:** 2
**Presentation:** 2
**Contribution:** 2
**Rating:** 4
**Confidence:** 5

**Summary:**

This paper studies **de novo peptide sequencing**, i.e., predicting peptide sequences directly from MS/MS spectra.
The authors propose **PLMNovo**, which introduces a **constraint-based learning** framework into the standard spectrum-to-sequence encoder–decoder architecture.
Specifically, they align **spectrum embeddings** with **peptide embeddings** generated by a **pretrained protein language model (PLM)** (ESM-2 8M/650M), and jointly train the model using a **Lagrangian primal–dual algorithm** (allowing optional LoRA fine-tuning of the PLM).
The PLM and alignment loss are used **only during training**, while inference still relies solely on the spectrum encoder and peptide decoder.
Experiments on a dataset of 2 million PSMs from **MSKB** show consistent improvements over **Casanovo 4.2** across both in-distribution (MSKB) and out-of-distribution (multi-enzyme) test sets, as well as on the **nine-species benchmark**.
The paper also analyzes how the alignment constraint shapes the joint embedding space and provides interpretability insights.

**Strengths:**

S1: The authors introduce a Protein Language Model (PLM) into the de novo peptide sequencing task, enhancing the model’s understanding of biological sequence knowledge.

S2: They employ a Lagrangian duality-based method to solve the sequencing problem under peptide–spectrum embedding alignment constraints.

**Weaknesses:**

W1: The idea of using a pretrained PLM as a peptide encoder combined with an alignment loss lacks novelty. It is conceptually similar to **ContraNovo (AAAI 2024)** [1], which also employs an additional peptide encoder and an alignment loss during training. Moreover, the authors did not include ContraNovo as a baseline for comparison in their experiments.

W2: In the ablation studies, the authors did not compare the performance of a **peptide encoder trained from scratch** with the same model size, which is necessary to justify that introducing a pretrained PLM indeed benefits this task

W3: Lines 295–296: “Treating the MSKB test set as a validation split.” Using the test set for hyperparameter tuning may inflate results on MSKB and compromises fairness when comparing to baselines. The authors should redefine the train/validation/test splits or use cross-validation.

W4: On the nine-species dataset, the model shows only marginal improvements at the amino-acid level, while its **peptide-level metrics (which are more important)** are consistently lower than Casanovo v2.

W5: The performance of PLMNovo (8M) and PLMNovo (650M) on the nine-species dataset is very similar, raising doubts about whether the knowledge encoded in larger PLMs actually provides meaningful gains for this task.

[1] ContraNovo: A Contrastive Learning Approach to Enhance De Novo Peptide Sequencing. AAAI2024

**Questions:**

Q1: **Comparison with ContraNovo:**
Since your approach of using a pretrained PLM as a peptide encoder with an alignment loss is similar to ContraNovo (AAAI 2024), why was ContraNovo not included as a baseline in your experiments? Could you clarify the main differences between PLMNovo and ContraNovo in terms of modeling strategy or optimization objective?

Q2:**Effect of pretrained PLM vs. training from scratch:**
Have you evaluated a version of PLMNovo where the peptide encoder is trained from scratch (with the same architecture and parameter size) rather than initialized from a PLM? Such a comparison is crucial to demonstrate that the pretrained PLM genuinely contributes to performance improvements.

Q3:**Data split and evaluation fairness:**
Lines 295–296 indicate that the MSKB test set was treated as a validation set for hyperparameter tuning. Could you provide results using a strictly held-out test split or a cross-validation protocol to ensure fair comparison with Casanovo and other baselines?

Q4:**Peptide-level performance degradation:**
On the nine-species dataset, PLMNovo improves amino acid–level metrics slightly but performs worse at the peptide level, which is arguably more important for biological interpretation. Can you provide analysis or insight into why the model struggles at the peptide level and how this might be addressed?

Q5:**Impact of PLM size and knowledge transfer:**
The results show that PLMNovo(8M) and PLMNovo(650M) achieve nearly identical performance on the nine-species dataset. Does this suggest that scaling the PLM or increasing its biological knowledge does not help the task? Have you examined whether the model actually leverages PLM-derived knowledge during training (e.g., through embedding similarity or representation analysis)?

---

### Official Review · Reviewer_t6eK · 2025-10-30

**Soundness:** 3
**Presentation:** 2
**Contribution:** 1
**Rating:** 2
**Confidence:** 5

**Summary:**

The paper introduces PLMNovo, a de novo peptide sequencing model that integrates pre-trained protein language models (PLMs) into a constrained optimization framework. It aligns embeddings from spectra and peptide sequences through a Lagrangian primal-dual training algorithm, encouraging biological consistency in the learned representations. The model reportedly improves amino acid precision and peptide recall across multiple datasets, including MSKB, multi-enzyme, and nine-species benchmarks, showing modest but consistent gains over Casanovo v4.2.

**Strengths:**

The main contribution is the novel formulation of peptide-spectrum alignment as a constrained optimization problem that leverages PLMs for biologically meaningful embeddings. The method is theoretically grounded with clear mathematical formalization and detailed training procedures. Experiments are extensive and include both in-distribution and out-of-distribution datasets, providing a comprehensive view of performance. The use of fine-tuning strategies such as LoRA is also a practical strength, showing attention to efficiency and scalability.

**Weaknesses:**

The paper overstates the significance of its improvements without sufficient quantitative justification, as most reported gains are marginal and possibly within statistical noise!!!! (Someone who worked with de novo sequencing for many years here, we tried the PLM for de novo, it never worked like it's showed in the paper.....THis looks like a failure study...... But I guess showing something not working can also benefit the community but plz submit to other workshop...... )

 The motivation for using constrained optimization rather than simpler regularization techniques is not clearly demonstrated through ablation or sensitivity analyses. The choice of the constraint bound ϵ and its effect on performance seem empirically tuned but not theoretically motivated.


 The dependency on ESM-2 embeddings limits generality since these PLMs are not inherently aware of post-translational modifications, which the authors acknowledge but do not attempt to mitigate. The constrained formulation adds complexity with dual variable updates and fine-tuning steps that could make training unstable or impractical for broader use, yet there is no discussion of convergence or runtime overhead. The experimental setup lacks proper comparison with newer non-autoregressive or diffusion-based models, which are current state-of-the-art directions in this domain. Overall, the paper presents a technically interesting idea but does not convincingly show that the added algorithmic complexity yields meaningful scientific or practical benefits.

**Questions:**

NA

---

### Note · Authors · 2025-11-22

I have read and agree with the venue's withdrawal policy on behalf of myself and my co-authors.